# Effect of Carrier-Transporting Layer on Blue Phosphorescent Organic Light-Emitting Diodes

Bo-Yen Lin [1,2], Chia-Hsun Chen [2,3], Tzu-Chan Lin [1], Jiun-Haw Lee [2,4] and Tien-Lung Chiu [1,*]

1    Department of Electrical Engineering, Yuan Ze University, Taoyuan 32003, Taiwan;
     boyenlin@saturn.yzu.edu.tw (B.-Y.L.); s1035603@mail.yzu.edu.tw (T.-C.L.)
2    Graduate Institute of Photonics and Optoelectronics, National Taiwan University, Taipei 10617, Taiwan;
     chiahsunchen@ntu.edu.tw (C.-H.C.); jiunhawlee@ntu.edu.tw (J.-H.L.)
3    Department of Chemistry, National Taiwan University, Taipei 10617, Taiwan
4    Department of Electrical Engineering, National Taiwan Univerity, Taipei 10617, Taiwan
*    Correspondence: tlchiu@saturn.yzu.edu.tw

**Abstract:** This study presented the effects of carrier-transporting layer (CTL) on electroluminescence (EL) performance of a blue phosphorescent organic light-emitting diodes (PHOLEDs) with electron transporting host based on three kinds of electron-transporting layers (ETLs) including 3-(4-biphenyl-yl)-4-phenyl-5-(4-tert-butylphenyl)-1,2,4-triazole (TAZ), diphenyl-bis[4-(pyridin-3-yl)phenyl]silane (DPPS) and 1,3,5-tri(m-pyrid-3-yl-phenyl)benzene (TmPyPB) and two kinds of hole-transporting layers (HTLs) such as 4,4′-bis[N-1-naphthyl-N-phenyl-amino]biphenyl (NPB), 1,1-bis[(di-4-tolylamino)phenyl]cyclohexane (TAPC). The carrier recombination and exciton formation zones in blue PHOLEDs strongly depend on the carrier mobility of CTLs and the layer thickness, especially the carrier mobility. Between ETLs and HTLs, the high electron mobility of ETL results in a lower driving voltage in blue PHOLEDs than the high hole mobility of HTL did. In addition, layer thickness modulation is an effective approach to precisely control carriers and restrict carriers within the EML and avoid a leakage emission of CTL. For CTL pairs in OLEDs using the electron transporting host system, ETLs with low mobility and also HTLs with high hole mobility are key points to confine the charge in EML for efficient photon emission. These findings show that appropriate CTL pairs and good layer thickness are essential for efficient OLEDs.

**Keywords:** phosphorescent organic light-emitting diodes; electron transporting; hole transporting





## 1. Introduction

Organic light-emitting diodes (OLEDs) consist of several organic stacks, such as hole-transporting layer (HTL), emitting layer (EML), and electron-transporting layer (ETL), which are sandwiched by electrodes [1–5]. At present, OLEDs are widely employed in flat panel and mobile phone as a screen since its attractive features such as light weight, thin size, high contrast ratio, fast response time, wide viewing angle, flexible, and more. Thanks to numerous efforts that have been invested in the development of material synthesis and device architectures in the past decade, the electroluminescence (EL) performance of OLEDs currently have achieved high efficiency and superior device stability for the requirement in commercialization, and the charge balance is considered as a crucial factor to determine the device performance [6–8]. ETLs and HTLs act important roles in charge of carrier injection, carrier transport, carrier and/or exciton confinement within EML for charge balance in OLEDs [9,10], which strongly depend on the charge transporting layer's (CTL's) photoelectric property including carrier mobility, energy band level (i.e., highest occupied molecular orbital (HOMO), lowest unoccupied molecular orbital (LUMO)) and the energy of singlet and triplet [11]. Although numerous efforts have been invested on the effects of ETLs and HTLs on device characteristics, such as device efficiency and operational lifetime, these papers reported the individual effect from either ETLs or HTLs.

For instance, Giebeler et al. investigated the effects of various HTLs on the device emissive characteristics [12]. In addition, Liu et al. reported the impact of ETLs on device stability under high current stressing [13], and indicated LUMO level and electron mobility as two other factors accounting for the degradation rate of a device. Most previous papers stressed on the study of the bipolar host system. However, some scarce papers stressed on electron transporting host system as well as investigated both the HTLs and ETLs simultaneously, and compared their effects on device characteristics to figure out the crucial parameters of them to determine a high-efficiency OLED device with a low driving voltage.

Here, the influence of CTLs on EL performance of a blue phosphorescent OLED (PHOLED) based on 3-(4-biphenyl-yl)-4-phenyl-5-(4-tert-butylphenyl)-1,2,4-triazole (TAZ) [14–17] doped bis[2-(4,6-difluorophenyl) pyridinato-C2, N] (picolinato)iridium(III) (FIrpic) as EML was investigated. Two HTLs, N, N-bis-(1-naphthyl)-N, N′-diphenyl-1,1′-biphenyl-4,4′diamine (NPB) [18,19], 1,1-Bis[(di-4-tolylamino) phenyl] cyclohexane (TAPC) [20,21] and three ETLs, TAZ, diphenyl-bis[4-(pyridin-3-yl)phenyl]silane (DPPS) [22–24]; and 1,3,5-tri(m -pyrid-3-yl-phenyl)benzene (TmPyPB) [25–28] were studied in this work. This is respective of TAZ, DPPS, and TmPyPB, whose electron mobility are approximately $\sim 10^{-5}$ (cm$^2$/Vs) [29], $\sim < 10^{-6}$ (cm$^2$/V s) [22], and $\sim 1 \times 10^{-3}$ (cm$^2$/Vs) [25]. Furthermore, hole mobility for NPB and TAPC are $5 \times 10^{-4}$ and $9.4 \times 10^{-3}$ (cm$^2$/Vs) [30,31], respectively. Various carrier mobility was applied to modify the carrier injection and carrier transport for observing their effects on the driving voltage, the efficiency, and the emissive spectrum of devices. In addition to the carrier mobility, the layer thickness of device structure acts an important role in charge of carrier transportation as well. Therefore, CTL thickness modulate is investigated.

## 2. Experiment

*OLED Fabrication and Measurement*

An indium-tin-oxide (ITO) coated glass substrate was used as anode and cleaned by detergent water, acetone and isopropyl alcohol (IPA) in sequence. Oxygen plasma treatment was used to raise the work function of ITO prior to thermal evaporation of organic thin-film stacks. After oxygen plasma treatment, a series of organic layers was deposited on ITO under a high vacuum condition of $\sim 10^{-6}$ torr, and then a 1.2 nm-thick lithium fluoride and a 100 nm-thick metallic cathode were deposited on organic stacks under a vacuum condition of $\sim 10^{-5}$ torr. For encapsulation, a glass with ultra-violet (UV) glue cover on the substrate under UV illumination for 12 min in a glove box. For device characterization, the luminance–current density–voltage (L–J–V) characteristics and EL spectrum were carried out using a source meter (Keithley 2400) and a spectrometer (Konica Minota CS-1000).

## 3. Results

*3.1. Electron Transporting Layer*

Figure 1 presents the schematic device structure of blue PHOLEDs and energy band diagram of materials as well as the chemical structure, where the values of HOMO and LUMO refer to the literature [20,30,32,33]. Table 1 shows the detailed device structure on the thickness of each organic layer and doping concentration of FIrpic in this work.

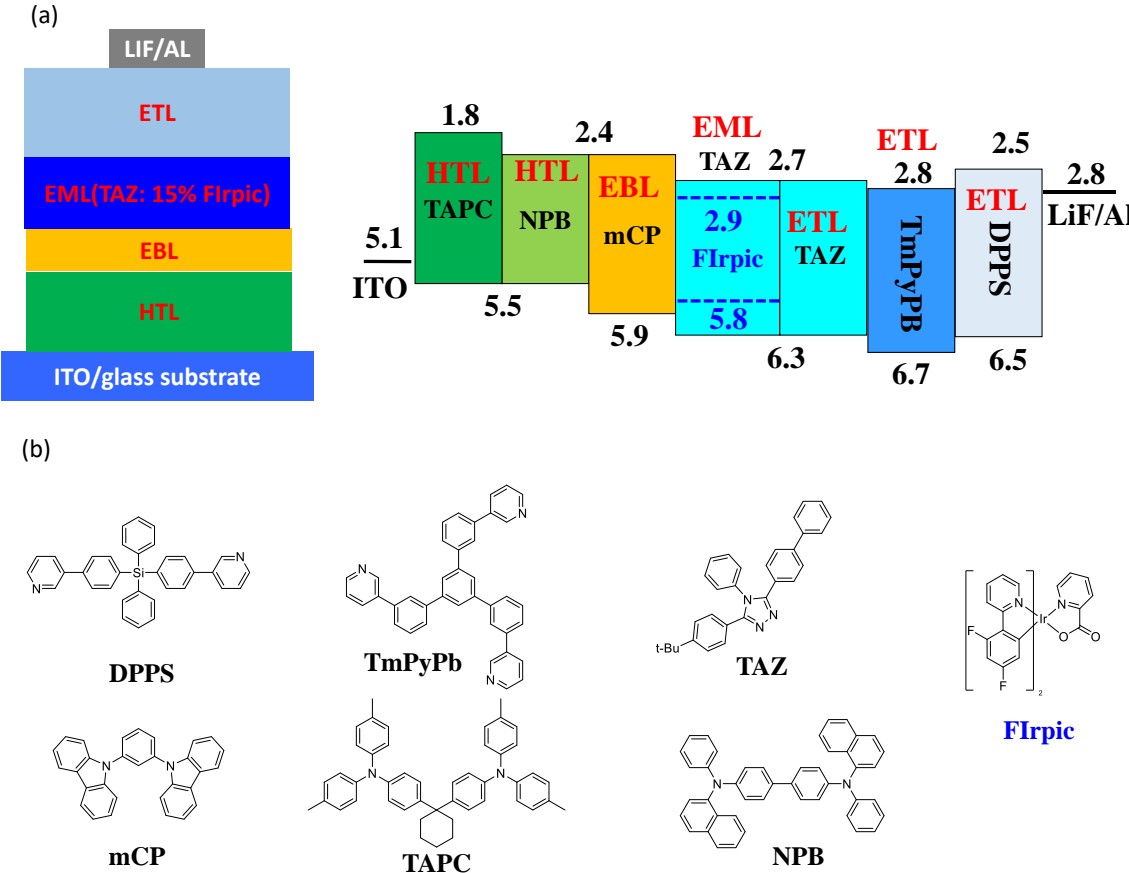

**Figure 1.** (**a**) Schematic device structure of blue phosphorescent organic light-emitting diodes (PHOLEDs) and energy band diagram of organic compounds of FIrpic, 3-(4-biphenyl-yl)-4-phenyl-5-(4-tert-butylphenyl)-1,2,4-triazole (TAZ), diphenyl-bis[4-(pyridin-3-yl)phenyl]silane (DPPS), diphenyl-bis[4-(pyridin-3-yl)phenyl]silane (DPPS) and 1,3,5-tri(m-pyrid-3-yl-phenyl)benzene (TmPyPB), mCP, 4,4′-bis[N-1-naphthyl-N-phenyl-amino]biphenyl (NPB) and 4,4′-bis[N-1-naphthyl-N-phenyl-amino]biphenyl (NPB), 1,1-bis[(di-4-tolylamino)phenyl]cyclohexane (TAPC), used in the blue PHOLEDs; (**b**) chemical structure of the organic compounds.

**Table 1.** Device structures for blue PHOLEDs with various electron-transporting layers (ETLs) and hole-transporting layers (HTLs).

| Device | HTL | EBL | EML | ETL |
|--------|-----|-----|-----|-----|
| A      |            |            |                        | TAZ (45 nm)    |
| B      |            |            |                        | TmPyPB (45 nm) |
| C      | NPB (50 nm)|            |                        | DPPS (45 nm)   |
| D      |            | mCP (10 nm)| TAZ: 15% Firpic (40 nm)| TAZ (40 nm)    |
| E      |            |            |                        | TAZ (50 nm)    |
| F      | TAPC (50 nm)|           |                        | TAZ 45 nm      |

Figure 2a shows the L–J–V characteristics of OLEDs with various ETLs. Three devices with the following structure: ITO/NPB (50 nm)/mCP (10 nm)/TAZ:FIpic (15% in volume) (40 nm)/TAZ (45 nm)/LiF (1.2 nm)/Al (100 nm), ITO/NPB (50 nm)/mCP (10 nm)/TAZ:FIpic (15% in volume) (40 nm)/TmPyPb (45 nm)/LiF (1.2 nm)/Al (100 nm) and ITO/NPB (50 nm)/mCP (10 nm)/TAZ:FIpic (15% in volume) (40 nm)/DPPS (45 nm)/LiF (1.2 nm)/Al (100 nm), were fabricated and they were denoted as device A, device B, device C. Since TAZ is known as an electron transporting host, the main carrier recombination zone (RZ) might be located at a position near to the HTL side in the EML. Here, NPB acts as HTL, which exhibited lower triplet energy of 2.3 eV than that of FIrpic, resulting in exciton quenching. mCP exhibiting $T_1$ = 2.9 eV is therefore inserted at HTL/EML interface

as an exciton-blocking layer (EBL) to avoid exciton quenching by NPB [34]. In addition, the electron injection barrier between EML and its adjacent ETL is small, which ranges only from 0 to 0.2 eV, and the difference of device characteristics mainly comes from the electron mobility. In Figure 2a, device B showed a more superior J–V characteristic than device A and C did. Compared with device A and C, device B exhibited the lowest driving voltage of 8.61 V at J = 20 mA/cm$^2$, which was due to the high electron mobility of TmPyPB (~$1 \times 10^{-3}$ cm$^2$/Vs). By contrast, since device A and device C have a low electron mobility of ETL (TAZ~$10^{-5}$ cm$^2$/Vs, DPPS < $10^{-6}$ cm$^2$/Vs), which exhibited the high driving voltage of 10.59 and 10.44 V, respectively. Figure 2b shows the current efficiency (CE) and external quantum efficiency (EQE) curves, where the maximum CE of device A, B and C are 45.93, 44.65 and 43.89 cd/A, respectively. Maximum EQE of Devices A, B, and C are 20.02%, 19.66% and 18.43%. Table 2 shows the summarized device performances. In addition, Figure 2c shows the EL spectra of device A, B and C measured at driving voltage of 12 V, and they correspond to CIE1931 coordinates of (0.188, 0.423), (0.214, 0.479), (0.196, 0.459), respectively. One found was a leakage emission from mCP occurred at a wavelength range of approximately 400–460 nm. For clarity, the inset of Figure 2c displays the EL spectra at the wavelength range 400–460 nm. Device A and B showed mCP emission (~430 nm), which significantly occurred in device B, due to the fact that the higher electron mobility of TmPyPB led to an efficient and smooth electron transport across the ETL to EBL. With respect to energy alignment to the corresponding ETL/EML, device A and B exhibited a well-matched energy level to EML compared to device C. Therefore, the electrons migrated difficulty across the ETL/EML interface, which retarded the electrons and avoided leaking to the mCP layer. Furthermore, these devices showed a main EL peak and two shoulder peaks of approximately 470 and 550 nm. The main EL peak was similar, but there was a different intensity in the shoulder EL peak of 550 nm. Devices A and B exhibited less shoulder emission than device C did, which was ascribed to the micro-cavity effect induced by different electron mobility of ETL generating the different width of recombination zone (RZ) [35,36]. A schematic model is illustrated in Figure 2d, where RZ is displayed with a varied width created by TAZ, TmPyPB, DPPS. The wider RZ in devices A and B produced a short optical length between RZ to ITO (Figure 2d) since the leakage electron expanded RZ to EBL, and the weak shoulder emissions were therefore obtained. By contrast, device C with DPPS showed narrow RZ and a strong shoulder emission due to the microcavity effect with a long optical length. According to aforementioned results, the leakage emission and the shoulder emission can clearly reflect the difference in electron mobility of ETLs. However, the leakage of mCP emissions represents an energy loss in blue PHOLEDs, so the ETL thickness modulation is a known way to control the electron migration. In the following section, ETL thickness was varied for investigation.

**Table 2.** Summary of OLEDs performances including driving voltage, luminance, CE and EQE.

| Device | Voltage [1] (V) | Luminance [2] (cd/m$^2$) | CE (cd/A) | EQE (%) |
|--------|-----------------|--------------------------|-----------|---------|
| A | 10.56 | 4441 | 45.93 [2], 25.95 [3] | 20.02 [2], 10.98 [3] |
| B | 8.61 | 5558 | 44.65 [2], 26.44 [3] | 19.66 [2], 13.22 [3] |
| C | 10.44 | 3393 | 43.89 [2], 25.60 [3] | 18.63 [2], 10.62 [3] |
| D | 9.72 | 5083 | 42.57 [2], 25.12 [3] | 17.41 [2], 9.98 [3] |
| E | 10.77 | 3952 | 45.66 [2], 25.82 [3] | 19.07 [2], 10.52 [3] |
| F | 10.34 | 4249 | 44.14 [2], 25.14 [3] | 17.77 [2], 9.77 [3] |

[1] 20 mA/cm$^2$, [2] Maximum, [3] L = 1000 cd/m$^2$.

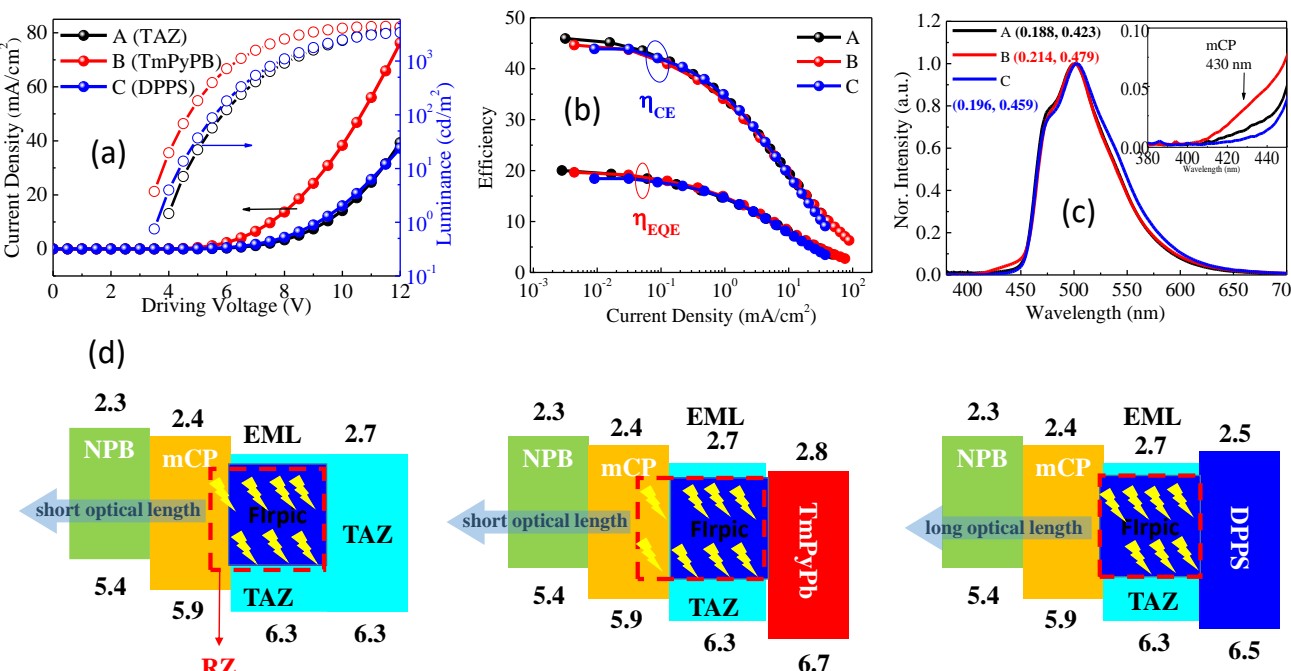

**Figure 2.** (**a**) L–J–V characteristics (**b**) current efficiency (CE) and external quantum efficiency (EQE) as a function of current density (**c**) EL spectra and color coordinates measured at driving voltage of 12 V; inset is EL spectra at the wavelength range 380–460 nm. (**d**) Schematic model in blue PHOLEDs with ETLs of TAZ, TmPyPb and DPPS.

### 3.2. Thickness of ETL

To understand the influence of ETL thickness, the ETL of device A, showing the highest performance of devices in Section 3.1, was thereby changed, decreased to 40 nm for device D and increased to 50 nm for device E. Figure 3 shows the L–J–V characteristics, CE and EQE as functions of current density and EL spectra of OLEDs with various TAZ thickness, ranging from 40–50 nm. While decreasing the thickness, although the lower driving voltage of 9.77 V was obtained in device D (Figure 3a), an efficiency drop also occurred simultaneously. CE and EQE decreased to 42.57 cd/A and 17.41% as shown in Figure 3b. In addition, as shown in Figure 3c, the leakage emission of mCP became obvious, which arises from that the thin TAZ thickness facilitated the electron migrating across the EML to mCP layer, and more electrons recombine with holes on mCP layer. This obvious leakage emission can explain why device D showed poor performance. On the other hand, device D with TAZ thickness of 50 nm exhibit similar EL performance as device A. As a consequence, the increase of TAZ thickness shows an insignificant effect on the elimination of mCP emission. Hence, another way to address this issue is employing an HTL exhibiting fast hole mobility.

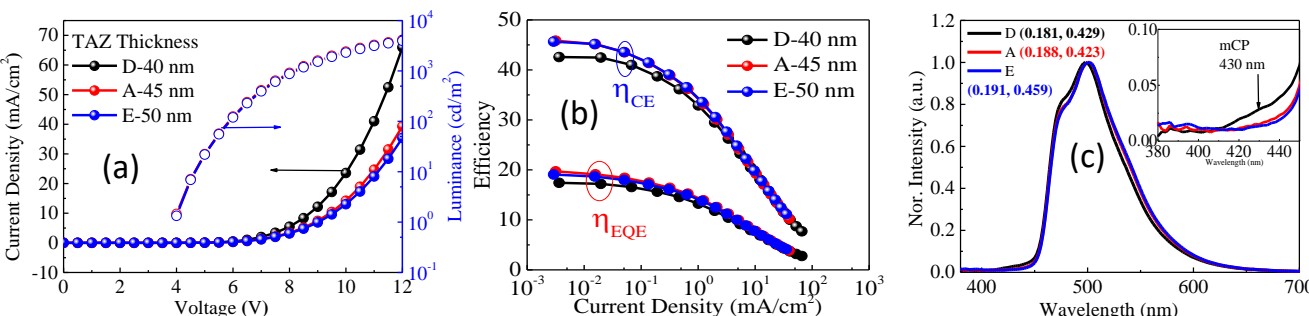

**Figure 3.** (**a**) L–J–V characteristics (**b**) CE and EQE as a function of current density (**c**) EL spectra and color coordinates measured at driving voltage of 12 V; inset is EL spectra at the wavelength range 380–460 nm, in blue PHOLEDs with various TAZ thickness.

### 3.3. Hole Transporting Layer

To realize the effects from HTL, TAPC with a hole mobility of $9.4 \times 10^{-3}$ $(cm^2/Vs)$ was employed to fabricate device F. The HOMO level of TAPC and NPB is the same, so the effect from the hole injection can be removed and the difference in device performance between these two HTLs is thereby in the hole mobility. Figure 4 shows the L–J–V characteristics of the OLEDs with various HTLs. Although hole mobility of TAPC is one order of magnitude higher than that of NPB, the driving voltage only slightly decreased from 10.56 to 10.34 V, and a small decrease of voltage was obtained, which indicated less driving voltage drop on the HTL layer in blue OLEDs. However, a large amount of voltage reduction was obtained in the ETL replacement, which indicated a higher voltage drop on ETLs in blue OLEDs. In efficiency, the CE decreased from 45.93 to 44.14 cd/A (Figure 4b) which might have been due to carrier unbalance (hole-rich). Figure 4c shows the spectra, where blue-shift in the emission color was observed in device F. The shoulder peaks of 470 nm were obvious due to RZ moving toward the cathode side. Similarly, this can be explained by the microcavity effect due to TAPC facilitating a superior hole transport, resulting in a wider RZ, extending to the TAZ layer, and a short optical length from RZ to the cathode as illustrated in Figure 4d [36]. Therefore, the mCP emission was eliminated by applying a faster HTL, but another CTL emission from TAZ occurred as is shown in the inset of Figure 4c.

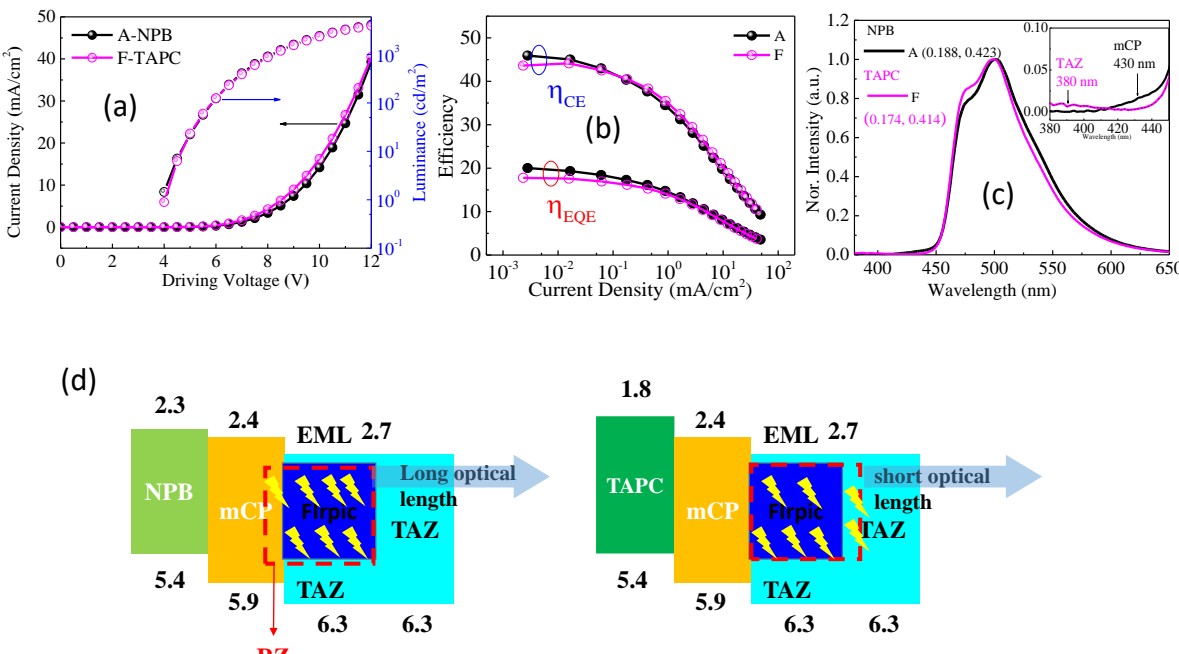

**Figure 4.** (**a**) L–J–V charateristics (**b**) CE and EQE as a function of current density (**c**) EL spectra and color coordinates measured at driving voltage of 12 V; inset is EL spectra at wavelangth range from 380 to 460 nm. (**d**) Schematic diagram of blue PHOLEDs with HTLs using NPB or TAPC.

### 4. Conclusions

In summary, the effects of ETLs and HTLs on EL performance of a blue PHOLED was conducted by materials, parameters and device architecture modulation including various carrier mobilities and CTL thickness variations. Both carrier mobility and layer thickness affect the driving voltage, especially the carrier mobility. Among ETLs and HTLs, the electron mobility of ETL dominates the driving voltage since the most driving voltage drop on ETL is in blue PHOLEDs. When applying CTLs exhibiting high carrier mobility in blue OLEDs, a significant voltage reduction can be obtained, but a worse carrier confinement and carrier balance in the EML and a leakage CTLs emission were also observed simultaneously, which resulted in an efficiency drop in the device. In addition, CTL thickness modulation can eliminate these issues for great efficiency performance.

Eventually, a conclusion for fabricating a blue PHOLED with electron transporting host was reached, which is that HTL and ETL pairs with high hole mobility and low electron mobility, respectively, are the key points to appropriately lead the charge confinement in EML for efficient photon emission. We believe that these findings offer design rules for a high-performance device.

**Author Contributions:** B.-Y.L.—Perform device fabrication and measurement and write this manuscript and revision. C.-H.C.—Perform device fabrication and measurement. T.-C.L.—Perform device fabrication and measurement. J.-H.L.—Instruct device design and measurement and revision. T.-L.C.—Instruct device design and measurement and revision. All authors have read and agreed to the published version of the manuscript.

**Funding:** This research received funding from Ministry of Science and Technology (MOST), Taiwan, under Grants MOST 109-2622-E-155-014, 108-2221-E-155-051-MY3, 108-2912-I-155-504, 108-2811-E-155-504, 107-2221-E-155-058-MY3, 107-2221-E-002-156-MY3, 107-2221-E-155-027, 107-3113-E-155-001-CC2, 106-3113-E-155-001-CC2, 106-2221-E-155-036, 106-2923-E-155-002-MY3, 106-2923-E-002-004-MY3, 105-2221-E-002-130-MY3. And The European Union's Horizon 2020 research and innovation program under the Marie Skłodowska-Curie grant agreement No 823720. And The APC of this paper was funded by MDPI.

**Institutional Review Board Statement:** Not applicable.

**Informed Consent Statement:** Not applicable.

**Data Availability Statement:** Not applicable.

**Acknowledgments:** This work was supported by the Ministry of Science and Technology (MOST), Taiwan, under Grants MOST 109-2622-E-155-014, 108-2221-E-155-051-MY3, 108-2912-I-155-504, 108-2811-E-155-504, 107-2221-E-155-058-MY3, 107-2221-E-002-156-MY3, 107-2221-E-155-027, 107-3113-E-155-001-CC2, 106-3113-E-155-001-CC2, 106-2221-E-155-036, 106-2923-E-155-002-MY3, 106-2923-E-002-004-MY3, 105-2221-E-002-130-MY3. JHL acknowledges support by the MEGA project, which has received funding from the European Union's Horizon 2020 research and innovation program under the Marie Skłodowska-Curie grant agreement No 823720.

**Conflicts of Interest:** There is no conflict of interest to declare.

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
