# Peer review of "Effect of Carrier-Transporting Layer on Blue Phosphorescent Organic Light-Emitting Diodes"

_photonics, doi:10.3390/photonics8040124_

Round 1
Reviewer 1 Report
In this work, the authors investigate the effects of carrier-transporting layers (both hole and electron ones) on the performance of blue phosphorescent OLEDs. Two HTLs and three ETLs with different carrier mobilities are employed. Both the mobility as well as the layer thickness affect the carrier recombination and exciton formation zone and, as a result, OLED performance. A higher electron mobility was found to lower the driving voltage more significantly than the corresponding hole mobility. Modulation of the layer thickness controlled carrier confinement and exciton formation in the active layer.
The study is well designed and nicely presented. Graphs and tables are highly appropriate. The reported data and the associated discussion provide some design rules concerning high perrformance of phosphorescent OLEDs. However, as the reported Materials have already been investigated previously, a comparison with data from the literature and a clearer message of how this study differs from other similar ones should be given. Furthermore, less atttention has been paid to the effect of the energy level alignment at the corresponding HTL (or ETL)/active layer interface in OLED performance. This is equally important and should not be ignored !
Author Response
Thanks to the reviewer’s helpful comments to rise up the quality of this paper to meet a high standard. Currently, in Introduction and Results of this draft were rewrote and modified to emphasize our innovation. In past decade, lots of efforts have been devoted to investigate influences of various ETLs and/or HTLs on device characteristics. Giebeler et al. investigated the effects of various HTLs on the device emissive characteristics [J. Appl. Phys. 1999, 85, 608–615]. Liu et al. reported the impact of ETLs on devices’ stability under high current stressing [Microelectron. Reliab. 2017, 71, 106–110.] Nevertheless, these papers focus on either ETLs’ effect or HTLs’ effect, but few papers focus on both ETLs and HTLs’ effect and paid attention to the concern about that which is the key factor to determine an efficient blue PHOLEDs among the ETLs and the HTLs. Furthermore, mostly previous papers studied the bipolar host system, but in our case electron transporting host is employed to investigated. Accordingly, the requirements of CTL is different than previous reports.
Therefore, to address above difference, we rewrote the Introduction with sentences “Although numerous efforts have been invested on the effects of ETLs and HTLs on device characteristics, such as device efficiency and operational lifetime, these papers reported the individual effect from either ETLs or HTLs. For instance, Giebeler et al. investigated the effects of various HTLs on the device emissive characteristics [11]. In addition, Liu et al. reported the impact of ETLs on devices’ stability under high current stressing [12], and indicated LUMO level and electron mobility as two other factors accounting for degradation rate of device. Most previous papers stressed on study of bipolar host system. However, rare papers stressed on electron transporting host system as well as investigated both the HTLs and ETLs simultaneously, and compared their effects on device characteristics to figure out the crucial parameters of them to determine a high-efficiency OLED device with a low driving voltage.” Moreover, in our case, an electron transporting host, TAZ, is employed in EML as a host, and the ETL materials including TAZ, TmPyPB, and DPPS are as adjacent ETLs. The electron injection barrier at EML/ETL interface ranged from 0 to 0.2 eV, which is in a reasonable range, and the ETLs’ effect on device characteristics supposedly depend on electron mobility. Nevertheless, the electron injection barrier is quietly small for energy level alignment, we still have added a short paragraph to discuss about this with following sentences “Respect to energy alignment to the corresponding ETL/EML, device A and B exhibited a well-matched energy level to EML than device C did. Therefore, the electrons difficultly migrated across the ETL/EML interface, that retarded the electrons and avoided leaking to mCP layer.” And also another paragraph was added in section 3.3 “To realize the effects from HTL, TAPC with a hole mobility of 9.4´10-3 (cm2/Vs) was employed to fabricated the device F. The HOMO level of TAPZ and NPB is the same, so the effect form the hole injection can be removed and the difference in device performance between these two HTLs is thereby in the hole mobility.”
Moreover, to address our contribution, we have summarized some key points for an efficient blue PHOLED with an electron transporting host. For CTL pairs choosing, ETLs should have a low mobility and also HTLs should have a high hole mobility, which are key points to confine charge in EML for efficient photon emission. The aforementioned conclusions are incorporated in the part of Abstract and Conclusion.
Revisions in the manuscript: To address the reviewer’s concern. The revised texts are incorporated into the Abstract on page 1 “For CTL pairs in OLEDs using electron transporting host system, ETLs with a low mobility and also HTLs with a high hole mobility are key points to confine charge in EML for efficient photon emission. These findings guide that an appropriate CTL pairs and well layer thickness is essential for an efficient OLEDs.” The revised texts are incorporated into the Introduction on page 1 & 2. “Although numerous efforts have been invested on the effects of ETLs and HTLs on device characteristics, ………………………………………………., and compared their effects on device characteristics to figure out the crucial parameters of them to determine a high-efficiency OLED device with a low driving voltage.” For energy level alignment, the revised texts are incorporated on page 4 & 5 with sentences “Respect to energy alignment to the corresponding ETL/EML, ……………., that retarded the electrons and avoided leaking to mCP layer.” Another revised texts are on page 5 with sentences “To realize the effects from HTL, TAPC with a hole mobility of 9.4´10-3 (cm2/Vs) was employed to fabricated the device F. The HOMO level of TAPZ and NPB is the same, so the effect form the hole injection can be removed and the difference in device performance between these two HTLs is thereby in the hole mobility.” The revised texts are incorporated into the Conclusions on page 7 “Eventually,………………………………. are the key points to lead well charge confinement in EML for efficient photon emission.”

Reviewer 2 Report
In this work, the authors investigated the effect of carrier-transporting layer on blue PHOLEDs. Three kinds of electron-transporting layers including TAZ, DPPS, and TmPyPB, and two kinds of hole-transporting layers including NPB and TAPC were used in building blue PHOLEDs. The authors that the carrier recombination and exciton formation zones in blue PHOLEDs strongly depend on the carrier mobility of CTLs and the layer thickness. The results of the paper are clearly supported by experimental data, making the manuscript's logic flow reasonable. The manuscript can be published after minor revision. Subsequent reviewing is necessary:
(1) In Figure 1, the chemical structure of NPB is obviously different from that of others. Please redraw the chemical structure of NPB.
(2) In the section of 3.1, the authors should summarize points that we can get from the difference of ETL
(3) We both know the CTL thickness can modulate the performance of OLEDs. The authors should give some design rules for fabricating blue PHOLEDs.
Reviewer 3 Report
The paper is interesting and novel. Unfortunately, the draft is written more like a technical than a scientific paper.
I advise authors that in the introduction, discuss more what was done in the literature and what is novel in their approach from previous work.
How carrier transport layer (physical and chemical properties) generally affects phosphorescent emission and, therefore, singlet-triplet dynamics.
Why their results are special and important.
Compare, for example, their approach, with work Urbina et al., who showed the effect of photonic confinement on phosphorescence emission, Molecular Physics, 2248, 2016
The discussion part should be significantly extended.
In the end, based on my previous comments, I am recommending a major revision.
Reviewer 4 Report
Dear Authors
You need to specify in the text what CTL is not just in the abstract. English sentences such as "We believe that these findings offer a design rules for a high performance device." needs to be corrected. For example here, a design needs to get rid of the "a".
Also, I think it will be wise to add a band bending analysis and/ or a device simulation to your results. For example there are software modules which can help you analyze this. If you can please do so.
Author Response
1st comment: You need to specify in the text what CTL is not just in the abstract. English sentences such as "We believe that these findings offer a design rules for a high performance device." needs to be corrected. For example here, a design needs to get rid of the "a".
Response: Thanks to the carefulness of the reviewer. The syntax error will be corrected in the edited draft. To address author’s concern, the error in the part of introduction have been revised, as indicated below “ETLs and HTLs act important roles in charge of carrier injection, carrier transport, carrier and/or exciton confinement within EML for charge balance in OLEDs [9,10], which strongly depend on the charge transporting layer’s (CTL’s) photoelectric property including carrier mobility, energy band level (i.e. highest occupied molecular orbital (HOMO), lowest unoccupied molecular orbital (LUMO)) and the energy of singlet and triplet.” Which are incorporated into the edited draft on page 1. The error in conclusion have been revised as well, the revised sentence was “We believe that these findings offer design rules for a high performance device.” Which are incorporated into the edited draft on page 7.
2nd comment: Also, I think it will be wise to add a band bending analysis and/ or a device simulation to your results. For example there are software modules which can help you analyze this. If you can please do so.
Response: Thanks to reviewer’s helpful suggestion. Unfortunately, we do not have these relative analysis tool.

Round 2
Reviewer 1 Report
My comments were satisfactorily addressed by the authors and, with the addition of some further discussion, now I find the manuscript suitable for publication.
Reviewer 3 Report
The authors provided correct answers to all referee questions, and the revised version reflects that.
Based on that fact, I am recommending the acceptance of the revised draft in unchanged form.